# Boosting Long-Context LLM Inference Efficiency with Intra-Layer Attention Similarity

## Abstract

The increasing context window size in Large Language Models (LLMs), such as the GPT and LLaMA series, has improved their ability to tackle complex, long-text tasks, but at the cost of inference efficiency, particularly regarding memory and computational complexity. Existing methods, including selective token retention and window-based attention, improve efficiency but risk discarding important tokens needed for future text generation. In this paper, we propose an approach that enhances LLM efficiency without token loss by reducing the memory and computational load of less important tokens, rather than discarding them. We address two challenges: 1) investigating the distribution of important tokens in the context, discovering recent tokens are more important than distant tokens in context, and 2) optimizing resources for distant tokens by sharing attention scores across layers. The experiments show that our method saves 35% KV cache without compromising the performance.

## 1 Introduction

Recently, the increasing context window size in Large Language Models (LLMs) (Brown et al., 2020; Achiam et al., 2023; Team et al., 2023; Reid et al., 2024; Touvron et al., 2023a;b; Dubey et al., 2024), has allowed them to excel in handling complex tasks necessitating an in-depth exploration of lengthy texts (Bairi et al., 2024; Mazumder & Liu, 2024). However, it poses challenges to the computation and memory footprint of LLMs. Specifically, on the one hand, since most LLMs are based on the Transformer (Vaswani et al., 2017) architecture, the computational complexity of the attention module increases quadratically with the size of the context window. On the other hand, the size of KV cache (Pope et al., 2023), a commonly used technique designed to prevent redundant computations, is linearly related to the context window size. Hence, enhancing the efficiency of LLMs with extended context windows is critical.

Against this backdrop, numerous researchers have put forward approaches to enhance the inference efficiency of LLMs by discarding some tokens within the context. In particular, the window attention approach (Beltagy et al., 2020) retains a fixed-size window over the KV states of the most recent tokens. LM-Infinite (Han et al., 2024) and StreamingLLM (Xiao et al., 2024) identify the "attention sink" phenomenon, preserving both the initial tokens and recent tokens (see Figure 1-(a)). $H_2O$ (Zhang et al., 2023) takes into consideration the differing significance of tokens within the context and selectively retains only the most important tokens in the KV cache based on attention scores. While such methods improve the efficiency of LLMs in handling long contexts, they introduce a major drawback (Tang et al., 2024): *critical tokens required for later text generation may be irreversibly discarded early in the process*. As shown in Figure 1-(a), when the important tokens (evidence in the example) fall outside the window, the prediction fails. Additionally, the performance degradation of StreamingLLM and $H_2O$ on two real-world benchmarks further confirms this (see Figure 1-(b)).

In this paper, we seek to improve the efficiency of LLMs while minimizing performance degradation. Our core motivation is that *less important tokens should be allocated fewer resources, rather than being discarded entirely*. This raises two challenges: 1) Where are the important tokens distributed for a token to attend, and 2) how to optimize memory and computation for less important tokens. We attempt to address these two challenges through two key observations:

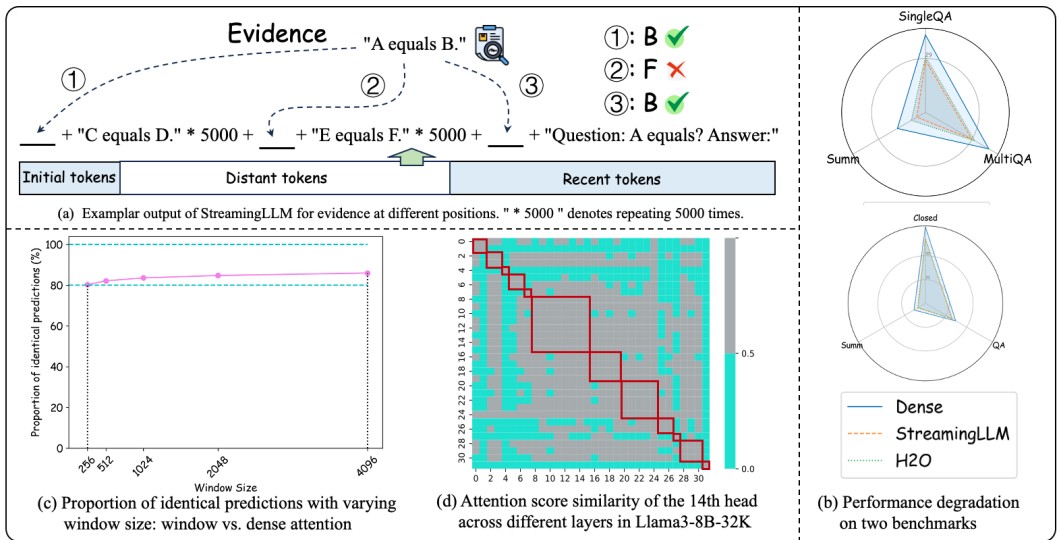

Figure 1: Experimental results from our customized LLaMA3-8B-32K model. (a) Example of StreamingLLM prediction failure. (b) Partial experimental results from LongBench (Bai et al., 2024) and LEval (An et al., 2024). (c) The test data consists of 1000 samples of 32K-length text sampled from Dolma (Soldaini et al., 2024). For each sample sequence, we compared the last 100 tokens. (d) We sampled 1000 sequences of 32K length from Dolma, extracting the attention scores for the last 16 tokens of each sample.

**Observation 1:** *Proximal tokens (initial tokens+recent tokens) are more important than distant tokens.* (see Figure 1-(c)). We conducted experiments and investigated the proportion of cases where the next-token predictions are identical when each token attends to only a fixed number of proximal tokens in the context, as opposed to attending to all tokens. Figure 1-(c) demonstrates that, even when attending to only the 256 proximal tokens, the model predicts the next token identically to the model attending to all tokens in 80% of the cases for the same input sequence. This phenomenon proves the observation 1.

**Observation 2:** *attention scores between consecutive layers are similar.* In fact, this phenomenon has been previously observed in smaller models (Xiao et al., 2019; Bhojanapalli et al., 2021), and here we scale it to modern LLMs. In Figure 1-(d), we can discover attention scores between the layers enclosed in the red box exhibit a strong similarity.

To this end, we propose PoD (Proximal tokens over Distant tokens) to optimize the inference efficiency during the *decoding* phase. In detail, it shares inter-layer attention scores exclusively for distant tokens, while leaving the proximal tokens unchanged based the above two observations. This approach consists of three main stages: 1) *Exploration of Offline Inter-Layer Attention Sharing* (§ 2.1): determining which layers can share attention scores; 2) *Lightweight Training Adaptation* (§ 2.2): post-training the dense model based on the identified attention sharing patterns between layers with a limited amount of data; 3) *Efficient Inference* (§ 2.3): sharing the attention scores between layers for distant tokens, which allows us to retain key states from a single layer in the KV cache. Additionally, we can preemptively identify situations where only proximal tokens are required to predict the next token, thereby eliminating the attention computation for distant tokens.

We evaluated the performance of PoD on Needle in a Haystack and two real-world long context benchmarks, analyzing its efficiency and examining the impact of key hyperparameters. Case studies were also conducted. Extensive experiments demonstrated that PoD can save 35% of the KV cache without compromising model performance. In summary, our contributions are: 1) we propose the idea of assessing the importance of tokens based on their positions in the context and enhancing inference efficiency by reducing the resources allocated to less important tokens. 2) we introduce PoD, a new model that adapts to the intra-layer shared attention distribution of distant tokens. 3) We conducted extensive experiments to prove PoD works and we plan to open-source our code and models in the future.

---

**Algorithm 1:** Greedy Layer Grouping Algorithm

---

**Input:** Head-wise attention similarities between layers: $\left\{ sim_h \left( \ell_a, \ell_b \right) \right\}_{1 \leq \ell_a, \ell_b \leq L}^{1 \leq h \leq H}$ and the
threshold $\delta$
**Output:** Head-wise layer blocks
head_wise_layer_blocks $\leftarrow []$;
**for** head $h \leftarrow 1$ *to* $H$ **do**
    current_head_layer_blocks$\leftarrow [\{1\}]$;              // Each block is a set.
    **for** layer $\ell \leftarrow 2$ *to* $L$ **do**
        current_block$\leftarrow$ the last element of current_head_layer_blocks;
        // Layer $\ell$ is similar to all layers in the current block.
        **if** $sim_h \left( \ell, \hat{\ell} \right) \geq \delta, \forall \hat{\ell} \in$ current_block **then**
            Add $\ell$ to current_block;
        **else**
            Append $\{\ell\}$ to current_head_layer_blocks;
    Append current_head_layer_blocks to head_wise_layer_blocks;
Return head_wise_layer_blocks;

---

## 2 METHODOLOGY

Our approach comprises three key steps (Figure 2): 1) analyze the similarity of attention scores between layers in a given long context LLM and group consecutive similar layers into blocks, 2) apply attention sharing within each block and post-train the LLM, and 3) conduct efficient inference by using the post-trained LLM.

### 2.1 EXPLORATION OF OFFLINE INTER-LAYER ATTENTION SHARING

To assess the similarity in attention scores between layers, we input several tokens into the LLM and collected a range of attention scores. Subsequently, We calculate the attention similarity between layers and group consecutive similar layers into blocks as a preparation for enhancing the inference efficiency of the LLM.

**Attention scores calculation** Assuming that we input $N$ samples $\left\{ \mathbf{s}_i = (x_1, x_2, \ldots, x_n) \right\}_{i=1}^{N}$ into the model $\mathcal{M}$, we will collect the attention scores of the last $q$ ($1 \leq q \leq n$) tokens attending to their corresponding previous tokens for each sample. Mathematically, we obtain

$$\left\{ \mathbf{S}_i^{\ell,h} \right\}_{1 \leq \ell \leq L, \, 1 \leq h \leq H} = \mathcal{M} \left( \mathbf{s}_i \right), \tag{1}$$

where $L, H \in \mathbb{N}^+$ denote the number of layers and attention heads in the model, respectively, and $\mathbf{S}_i^{\ell,h} \in \mathbb{R}^{q \times n}$ represents the attention scores collected at the $\ell$-th layer of the $h$-th attention head.

**Attention similarity evaluation** Next, we need to evaluate the similarity between layers based on the collected attention scores. For any two distinct layers $\ell_a$ and $\ell_b$ ($1 \leq \ell_a, \ell_b \leq L$ and $\ell_a \neq \ell_b$), the attention similarity between them for the $h$-th head is defined as the average Jensen-Shannon (JS) divergence (Menéndez et al., 1997) over the last $q$ tokens across all $N$ samples. Formally,

$$sim_h \left( \ell_a, \ell_b \right) = \frac{1}{N \cdot q} \sum_{i=1}^{N} \sum_{j=1}^{q} \mathrm{JS} \left( \mathbf{S}_{i,j}^{\ell_a,h}, \mathbf{S}_{i,j}^{\ell_b,h} \right), \tag{2}$$

where $\mathbf{S}_{i,j}^{\ell,h}$ denotes the $j$-th row of $\mathbf{S}_i^{\ell,h}$ and $0 \leq sim_h \left( \cdot, \cdot \right) \leq 1$.

**Layer grouping** After calculating the head-wise attention similarity between layers, we group consecutive similar layers into head-wise blocks in preparation. Our grouping strategy is based on the idea that any two layers within the same block should be sufficiently similar. Elaborately, $\ell_a$ and

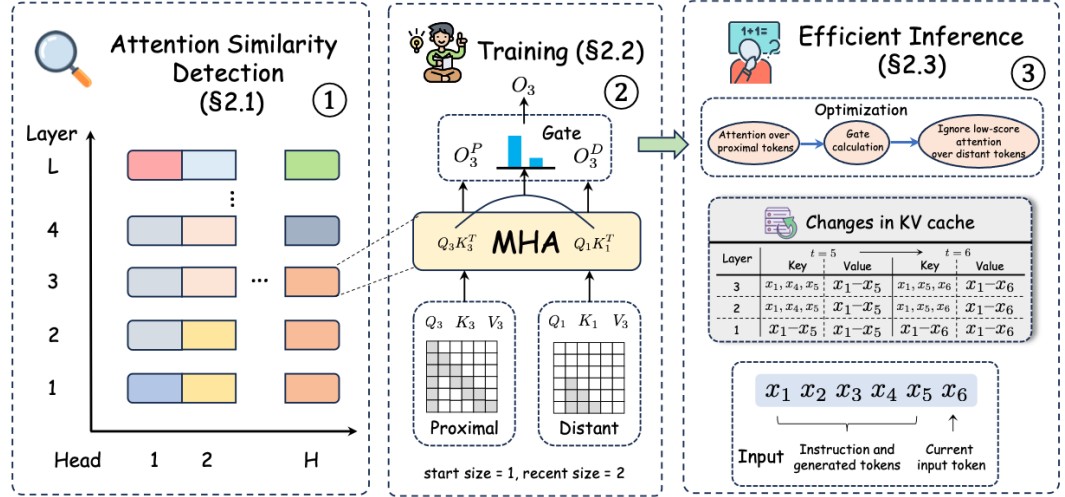

Figure 2: Three stages of PoD. ①: An example of layer grouping based on the similarity of offline-computed attention scores for each head. Consecutive layers with the same color belong to the same block and thus utilize the same attention scores. ②: Exemplar model structure of the attention module for head 3 in PoD. Each token attends to two groups: proximal tokens (the neighboring $n_r$ tokens and the initial $n_s$ tokens, $n_r = 2$ and $n_s = 1$ in this example) and distant tokens (the remaining tokens). Attention is computed separately for both groups and weighted based on their respective attention scores. ③: Example of KV cache changes for the 7-th token and avoidance of distant token attention.

$\ell_b$ are considered similar when $sim(\ell_a, \ell_b) \geq \delta$, where $0 \leq \delta \leq 1$ is a hyperparameter. Building on this, we adopt a bottom-up greedy algorithm to iteratively merge consecutive similar layers into blocks, as detailed in Algorithm 1. Figure 2-① presents a simple example of head-wise partitioning of layers based on the similarity of inter-layer attention scores.

## 2.2 LIGHTWEIGHT TRAINING ADAPTATION

To enhance the capability of the model to share attention across layers within the same block, we first introduce the post-training process required to adapt the model to this mechanism.

**Attention sharing within each block** We denote the long input sequence as $\mathbf{s} = (x_1, x_2, \ldots, x_n)$. For any token $x_i$ ($1 \leq i \leq n$) at the $\ell$-th layer in autoregressive transformer-based LLMs, it attends to all preceding tokens $\{x_j\}_{j \leq i}$. As mentioned, we aim to optimize memory and computation costs for distant tokens by sharing attention layers. Hence, we first categorize the preceding tokens into two groups: proximal tokens and distant tokens. Following previous works (Han et al., 2024; Xiao et al., 2024), we classify several initial tokens as proximal tokens, taking into account the phenomenon of the "attention sink". Then, token $x_i$ will attend to both groups of tokens, but the attention scores for distant tokens will utilize the attention scores from the lowest layer of the corresponding block grouped in Section 2.1.

Mathematically, for any attention head, let $\mathbf{Q}_\ell, \mathbf{K}_\ell, \mathbf{V}_\ell \in \mathbb{R}^{n \times d}$ denote the query, key, and value states at the $\ell$-th layer, respectively[*]. Layer $\ell$ belongs to the block $B_\ell = \{\bar{\ell} \mid \ell_a \leq \bar{\ell} \leq \ell_b\}$, which consists of consecutive layers. The outputs of attention for the proximal tokens and the distant tokens

---

[*]For the sake of simplicity, we omit the subscripts for the attention heads here.

for the token $x_i$ are calculated as follows[†]:

$$\mathbf{a}_{\ell,i}^P = \frac{\mathbf{Q}_{\ell,i}\left[\mathbf{K}_{\ell,[1,n_s]};\mathbf{K}_{\ell,[n-n_r+1,n]}\right]^T}{\sqrt{d}}, \quad \mathbf{o}_{\ell,i}^P = \text{Softmax}\left(\mathbf{a}_{\ell,i}^P\right)\left[\mathbf{V}_{\ell,[1,n_s]};\mathbf{V}_{\ell,[n-n_r+1,n]}\right],$$

$$\mathbf{a}_{\ell,i}^D = \frac{\mathbf{Q}_{\ell_a,i}\mathbf{K}_{\ell_a,[n_s+1,n-n_r]}^T}{\sqrt{d}}, \qquad\qquad \mathbf{o}_{\ell,i}^D = \text{Softmax}\left(\mathbf{a}_{\ell,i}^D\right)\mathbf{V}_{\ell,[n_s+1,n-n_r]}, \tag{3}$$

where $\mathbf{Q}_{\ell,i}$ represents the $i$-th row of $\mathbf{Q}_\ell$, $\mathbf{K}_{\ell,[a,b]}$ denotes the rows from the $i$-th to the $j$-th row of $\mathbf{K}_\ell$, inclusive of the boundaries. Additionally, $n_s$ (start size) and $n_r$ (recent size) represent the number of initial and recent tokens within the proximal tokens, respectively. $[\cdot;\cdot]$ is the concatenation operation. $\mathbf{a}_{\ell,i}^P \in \mathbb{R}^{1\times(n_s+n_r)}$ is the attention inner product to the proximal tokens and $\mathbf{o}_{\ell,i}^P \in \mathbb{R}^{1\times d}$ is the attention output to the proximal tokens for the token $x_i$. Notations for attention to the distant tokens are similar.

**Aggregation of attention output to proximal and distant tokens**    A parameter-free gating mechanism can integrate attention to proximal and distant tokens via[‡]

$$g_{\ell,i} = \frac{\sum \exp \mathbf{a}_{\ell,i}^P}{\sum \exp \mathbf{a}_{\ell,i}^P + \sum \exp \mathbf{a}_{\ell,i}^D}, \qquad \mathbf{o}_{\ell,i} = g_{\ell,i}\cdot\mathbf{o}_{\ell,i}^P + (1-g_{\ell,i})\cdot\mathbf{o}_{\ell,i}^D. \tag{4}$$

Figure 2-② is an example of parallel training with an attention mask.

## 2.3 EFFICIENT INFERENCE

Next, we will discuss strategies for optimizing memory usage and reducing computations for distant tokens in long context large language model inference through the use of layer-sharing in attention mechanisms.

**KV cache memory footprint optimization**    As illustrated in Equation 3, the query and key states are shared across layers within the same block for distant tokens. During inference, caching query states is unnecessary, as they are not reused. Consequently, our method will reduce the memory consumption of key states in the KV cache. Figure 2-③ presents an example of KV cache changes. In this case, the layers that share attention scores during decoding retain distant tokens only once; for instance, only layer 1 retains the key states for $x_2$ and $x_3$, while layers 2 and 3 do not.

**Computation optimization for distant tokens**    Empirical evidence suggests that in many situations, the prediction of the next token can be effectively accomplished without attending to distant tokens. This is reflected in Equation 4, where $g_{\ell,i}$ approaches 1 in numerous cases. Based on this, as shown in the top of Figure 2-③, for layers within a block that are not the lowest, we can preemptively evaluate the value of $g_{\ell,i}$. If $g_{\ell,i} \geq \tau$ ($0 \leq \tau \leq 1$ is a hyperparameter), the computation of attention for distant tokens can be omitted, thereby reducing computation for distant tokens.

## 3 EXPERIMENTS

In this section, we explore two key questions: 1) whether PoD experiences performance degradation, and 2) whether PoD improves efficiency in long-context inference.

**Implementation details**    For the data, we sampled a total of 5B tokens from Dolma (Soldaini et al., 2024) for post-training, ensuring that the total number of tokens in each length interval remains consistent (GLM et al., 2024). For the model, we first initialized using LLaMA3-8B and conducted post-training on the 5B tokens with a maximum sequence length of 32K, resulting in LLaMA3-8B-32K. Subsequently, we initialized from LLaMA3-8B-32K and continued post-training on the same 5B tokens with a maximum sequence length of 32K, yielding the PoD model with $n_s = 16$ and $n_r = 4080$. The layer similarity threshold $\delta$ is set to 0.5, corresponding to saving 35% KV cache

---

[†]When there are no distant tokens for $x_i$, attention to distant tokens does not exist.

[‡]The derivation process for calculating the gate is in the Appendix A.

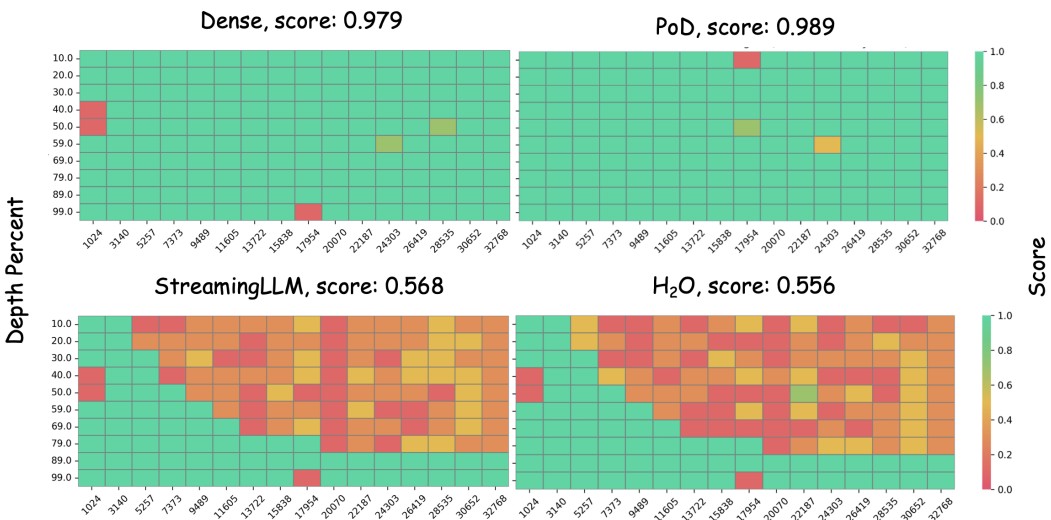

Figure 3: Searching results for a needle in a haystack

states. During training, the batch size was set to 4M tokens, with a learning rate of 1e-5 and a cosine learning rate scheduler. Additionally, the base in the RoPE (Rotary Positional Embedding) (Su et al., 2023) was increased to 16M+, as in Xiong et al. (2024). For the code implementation, we utilized HuggingFace (Wolf et al., 2020) along with DeepSpeed (Rasley et al., 2020), incorporating ZeRO-3 (Rajbhandari et al., 2020) and Ulysses (Jacobs et al., 2023) sequence parallelism techniques, and employed efficient FlexAttention from PyTorch (Paszke et al., 2019).

**Baselines**    We primarily consider the following three types of baselines:

- *Token-selection-based methods*: 1) SnapKV (Li et al., 2024): important tokens are selected based on attention scores, and only these important tokens are cached during the prefilling phase. 2) PyramidKV (Zhang et al., 2024b): this work extends SnapKV, where the number of tokens cached at different layers varies. 3) Quest (Tang et al., 2024): this work does not reduce the size of the KV cache, but instead reduces the number of tokens involved in the attention computation that are read from the KV cache through an efficient token selection method.

- *Token-eviction-based methods*: 1) Window Attention (WA) (Beltagy et al., 2020): each token will only attend to a limited number of neighboring tokens. 2) Window Attention + CPT (Continual Pre-Training): continue pre-training LLaMA3-8B-32K on the same 5B tokens with window attention. 3) StreamingLLM (Xiao et al., 2024): in addition to neighboring tokens, each token will also attend to the initial few tokens. 4) LM-Infinite (Han et al., 2024): each token attends to the same tokens as in StreamingLLM, but the position embeddings differ. 5) $H_2O$: each token not only focuses on neighboring tokens but also dynamically adds important tokens and removes less significant tokens based on the attention scores during decoding.

- *Layer-sharing-based methods*: CLA (Brandon et al., 2024) reduces the KV cache by sharing key and value states across adjacent layers.

### 3.1 PERFORMANCE EVALUATION

To evaluate the performance of PoD, we conducted experiments in two fields: 1) Needle in a Haystack and 2) Practical Long Context Benchmarks.

**Needle in a Haystack**    The task places a random statement in the middle of a long context window and asks the model to retrieve this statement. Figure 3 demonstrates the searching results of different methods. We found that StreamingLLM and $H_2O$ fail when the needle is outside their predefined

Table 1: Evaluation results of different methods on two famous long context benchmarks

| Model | Window | LongBench | | | | | | LEval | | | |
| | | SQA | MQA | Summ | Few-Shot | Code | Avg. | Closed | QA | Summ | Avg. |
|---|---|---|---|---|---|---|---|---|---|---|---|
| LLaMA3-8B-32K | 32K | 32.94 | 32.23 | 25.41 | 69.30 | 66.54 | 45.28 | 42.10 | 24.68 | 15.55 | 27.45 |
| *Token-selection-based methods* | | | | | | | | | | | |
| SnapKV | 4K | 31.76 | 31.85 | 21.92 | 68.62 | 66.72 | 44.17 | 39.86 | 23.90 | 13.53 | 25.76 |
| PyramidKV | 4K | 33.34 | 31.51 | 23.76 | 68.91 | 66.36 | 44.78 | 42.10 | 22.63 | 12.96 | 25.90 |
| Quest | 4K | 32.14 | 32.19 | 24.27 | 69.05 | 66.43 | 44.82 | 40.55 | 25.59 | 14.69 | 26.94 |
| *Token-eviction-based methods* | | | | | | | | | | | |
| LM-Infite | 16+4080 | 28.83 | 28.95 | 21.74 | 68.12 | 66.54 | 42.84 | 37.32 | 22.80 | 13.91 | 24.68 |
| StreamingLLM | 16+4080 | 28.68 | 28.95 | 21.64 | 68.14 | 66.60 | 42.80 | 37.12 | 22.79 | 13.81 | 24.57 |
| $H_2O$ | 96+4000 | 29.36 | 29.51 | 22.73 | 68.45 | 66.17 | 43.24 | 37.15 | 23.18 | 13.48 | 24.60 |
| WA | 4K | 8.90 | 3.63 | 9.05 | 11.13 | 41.08 | 14.76 | 20.95 | 5.57 | 2.79 | 9.77 |
| WA + CPT | 4K | 26.94 | 27.95 | 22.29 | 66.60 | 66.10 | 41.97 | 32.94 | 22.09 | 12.55 | 22.52 |
| *Layer-sharing-based methods* | | | | | | | | | | | |
| CLA | 32K | 24.02 | 22.58 | 22.50 | 60.92 | 59.35 | 37.87 | 19.05 | 13.52 | 11.52 | 14.70 |
| PoD (ours) | 16+4080+28K | 30.97 | 32.43 | 24.82 | 67.30 | 68.26 | 44.75 | 43.59 | 22.95 | 15.00 | 27.18 |
| PoD+SnapKV (ours) | 4K | 30.98 | 32.68 | 22.90 | 66.90 | 67.79 | 44.25 | 43.07 | 22.12 | 14.32 | 26.50 |

window. In contrast, our method, which avoids token loss, performs similarly to dense models and can locate nearly all the needles.

**Long Context Benchmarks** To ensure that PoD can handle real-world tasks, we evaluated it on two well-known long context benchmarks: LongBench (English version) (Bai et al., 2024) and LEval (An et al., 2024). We test on 14 datasets within LongBench involving Single-document QA, Multi-document QA, Summarization, Few-shot learning, and Code completion tasks. LEval consists of 20 sub-tasks, divided into two groups: closed-domain and open-domain. The closed-domain group primarily evaluates reasoning and comprehension over longer contexts, while the open-domain group focuses on tasks such as summarization and question answering, which require aggregating information from long documents.

Table 1 illustrates all experimental results. To ensure fairness, all baseline attention mechanisms have the same window size. For PoD, we also ensure that the number of proximal tokens each token attends to is consistent with this window size.

We can draw the following conclusions: 1) PoD outperforms token-eviction-based methods, demonstrating that our approach of not losing tokens is indeed effective. 2) With a small amount of post-training data, PoD beats the classical layer-sharing-based method CLA, demonstrating that our model has an advantage in adapting existing LLMs. 3) Both PoD and token-selection-based methods can achieve performance comparable to the standard dense model. Furthermore, PoD is *orthogonal* to token-selection-based methods, and combining them can further reduce the size of the KV cache while maintaining model performance.

## 3.2 EFFICIENCY EVALUATION

**Memory footprint** The savings in memory consumption can be analyzed from both theoretical and empirical perspectives. Theoretically, we can calculate the potential reduction in KV cache size based on the layer-sharing results obtained from offline analysis. Empirically, we can conduct end-to-end evaluations to assess the actual savings. Following Flex-Gen (Sheng et al., 2023) and LCKV (Wu & Tu, 2024), for a prompt of length $x$, we let the model generate $y$ tokens, The maximum batch size $b$ achievable on a given GPU will be used to assess the memory requirements of the model. A larger $b$ indicates that the model is more memory-efficient. Table 2 presents the memory consumption results. We observe that PoD achieves a more than 30% increase in maximum batch size across varying input text lengths, closely aligning with our theoretical KV cache savings rate of 35%, demonstrating that PoD effectively reduces memory usage.

Table 2: Theoretical and practical memory footprint savings

| Theoretical saving | Practical evaluation of maximum batch size $b$ | | | |
| | $x$ | $y$ | LLaMA3-8B-32K | PoD |
|---|---|---|---|---|
| 35% | 2048 | 8192 | 25 | 33 (32.0% ↑) |
| | 4096 | 8192 | 13 | 17 (30.8% ↑) |
| | 8192 | 8192 | 6 | 8 (33.3% ↑) |
| | 16384 | 8192 | 3 | 4 (33.3% ↑) |

**Optimized stage, KV cache saving, and performance balance** In Table 3, we compare different methods based on the optimized stage (prefilling or decoding), the proportion of KV cache saved, and the resulting performance loss. From this, we can draw the following conclusions: 1) Token-eviction-based methods are more efficient, as they are suitable for both prefilling and decoding and save a relatively large amount of KV cache. However, they come with a greater performance loss for the model. 2) Token-selection-based methods (SnapKV & PyramidKV) can significantly compress the KV cache with minimal impact on model perfor-

Table 3: Optimized stage, KV cache saving, and performance balance between different methods

| Method | Optimized Stage | KV Cache Saving (%) | Performance Degradation (%) |
|---|---|---|---|
| *Token-selection-based methods* | | | |
| SnapKV | Prefilling | 87.5 | 4.3 |
| PyramidKV | | 93.6 | 3.4 |
| Quest | Prefilling&Decoding | 0.0 | 1.4 |
| *Token-eviction-based methods* | | | |
| LM-Infinite | | | 7.7 |
| StreamingLLM | | | 8.0 |
| $H_2O$ | Prefilling&Decoding | 87.5 | 7.4 |
| WA | | | 65.9 |
| WA+CPT | | | 12.6 |
| *Layer-sharing-based methods* | | | |
| CLA | | 50.0 | 31.4 |
| PoD (ours) | Prefilling&Decoding | 35.0 | 2.8 |
| PoD+SnapKV (ours) | | 91.9 | 3.1 |

mance, but they are only applicable to the prefilling stage. Quest, on the other hand, does not degrade model performance and can be applied to both prefilling and decoding stages. However, it does not reduce the size of the KV cache; instead, it reduces the KV cache load by limiting the model's attention to a selectively filtered subset of tokens. 3) Layer-selection-based methods can be applied to both the prefilling and decoding stages. Our method incurs much less performance degradation compared to the classical layer-sharing method, CLA. Furthermore, our method is orthogonal to token-selection-based methods, and when the two are combined, the resulting model excels in three aspects: optimized stage (applicable to both prefilling and decoding), KV cache saving (with a higher compression rate), and maintaining model performance.

**Computation for distant tokens** As is mentioned in § 2.3, the attention output of a particular layer, is derived from the weighted outputs of both proximal and distant tokens. The weighting coefficients can be obtained in advance using the shared attention scores from the lower layer before applying attention to the distant tokens. Therefore, when the weighting coefficient for the proximal tokens is greater than $\tau$, we consider the distant tokens to be irrelevant for predicting the next token at the current decoding time in this layer, allowing us to skip their computation.

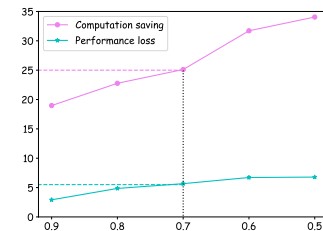

Figure 4: The computation saving and performance loss rates versus the gate threshold $\tau$ mentioned in § 2.3.

Figure 4 shows the relationship between the ratio of computational savings and performance loss on LEval and the value of $\tau$. We observe that as $\tau$ decreases, it becomes easier to ignore the computation for distant tokens, leading to greater computational savings, but with some performance loss. However, when $\tau < 0.7$, the performance degradation slows down while the computational savings become more pronounced. Specifically, when $\tau = 0.7$, a 25% reduction in computational cost is achieved with only a 5% decrease in performance.

## 3.3 ADDITIONAL ANALYSIS

**Scaling to longer context and other LLMs** To explore the generality of our method, we conducted experiments on LLaMA3.1-8B (Dubey et al., 2024), which can handle longer (128K) contexts. We sampled 5B tokens from the ProLong-data-512K (Gao et al., 2024) dataset and applied the same hyperparameter configuration used for training LLaMA3-8B-32K to post-train LLaMA3.1-8B with a sequence length of 128K. The evaluation results over 4 practical sub-tasks in the InfiniteBench (Zhang et al., 2024a) under different context sizes are shown in Table 4.

Consistent with the conclusions found in Table 1, our method causes less performance degradation compared to token-eviction-based methods. However, a notable difference is that token-selection-based methods appear to struggle in maintaining model performance in longer context scenarios. This limitation is also reflected in the combined model (PoD+SnapKV), which integrates our

Table 4: Evaluation results on InfiniteBench. OOM: out of memory over one A800-80G GPU

| Model | Window | 32K | | | | 64K | | | | 128K | | | | Avg. |
|---|---|---|---|---|---|---|---|---|---|---|---|---|---|---|
| | | En.MC | En.Summ | Math.Find | Code.Debug | En.MC | En.Summ | Math.Find | Code.Debug | En.MC | En.Summ | Math.Find | Code.Debug | |
| LLaMA3.1-8B | 128K | 25.33 | 15.51 | 32.29 | 27.41 | 28.82 | 15.14 | 28.00 | 27.92 | 41.48 | 14.86 | 21.14 | 25.38 | 25.27 |
| *Token-selection-based methods* | | | | | | | | | | | | | | |
| SnapKV | 4K | 26.64 | 13.26 | 32.28 | 27.41 | 27.07 | 11.69 | 28.00 | 27.92 | 34.50 | 13.65 | 21.05 | 26.65 | 24.18 |
| PyramidKV | 4K | 25.76 | 15.01 | 32.29 | 27.39 | 28.82 | 15.18 | 28.00 | 27.92 | OOM* | | | | OOM |
| Quest | 4K | 25.76 | 12.87 | 32.29 | 27.41 | 27.51 | 11.22 | 27.89 | 27.90 | 34.50 | 8.95 | 21.14 | 26.9 | 23.70 |
| *Token-eviction-based methods* | | | | | | | | | | | | | | |
| LM-Infite | 16+4080 | 26.64 | 12.46 | 32.11 | 27.31 | 28.82 | 12.64 | 26.89 | 27.91 | 29.26 | 13.01 | 21.14 | 26.63 | 23.74 |
| StreamingLLM | 16+4080 | 26.20 | 12.05 | 32.29 | 27.41 | 27.95 | 13.29 | 28.00 | 27.92 | 27.95 | 12.79 | 21.14 | 26.65 | 23.64 |
| H2O | 96+4000 | 26.20 | 13.08 | 32.29 | 27.41 | 27.95 | 14.31 | 28.00 | 27.91 | OOM* | | | | OOM |
| WA | 4K | 3.49 | 0.53 | 0.00 | 10.41 | 3.06 | 0.65 | 0.00 | 10.66 | 3.49 | 0.70 | 0.00 | 9.64 | 3.55 |
| WA + CPT | 4K | 36.79 | 11.31 | 18.57 | 28.43 | 38.47 | 10.78 | 17.71 | 28.68 | 39.36 | 11.26 | 17.71 | 28.17 | 23.94 |
| *Layer-sharing-based methods* | | | | | | | | | | | | | | |
| CLA | 32K | 34.06 | 13.58 | 12.57 | 27.16 | 31.88 | 12.74 | 12.57 | 27.66 | 34.50 | 12.96 | 12.86 | 29.44 | 21.83 |
| PoD (ours) | 16+4080+28K | 33.12 | 16.42 | 24.71 | 26.65 | 37.17 | 15.60 | 21.71 | 26.14 | 40.61 | 15.05 | 22.29 | 25.89 | 25.45 |
| PoD+SnapKV (ours) | 4K | 29.26 | 12.89 | 27.71 | 26.65 | 37.12 | 12.56 | 21.71 | 26.14 | 31.00 | 13.23 | 22.29 | 25.89 | 23.87 |

method with token-selection-based methods, showing a decline in performance. This to some extent indicates that our method is more robust to the context length.

Next, we explore two key hyperparameters in POD: the number of proximal tokens and the degree of attention score sharing between layers, which is reflected in the KV cache savings rate. Starting from the LLaMA3-8B-32K initialization, we continued training with 2B data to conduct experiments.

**Relationship between model performance and numbers of proximal tokens** As shown in the left part of Figure 5, the performance of POD steadily improves with an increasing number of proximal tokens. When the count reaches 4K tokens, training with 2B data achieves performance that is acceptable compared to the LLaMA3-8B-32K trained with 5B data. Considering the trade-off between performance and efficiency, we ultimately chose to use 4K proximal tokens.

**Relationship between model performance and KV cache savings rate** As shown in the right part of Figure 5, the performance of POD decreases as the KV cache savings rate increases. Considering the balance between performance and efficiency, we ultimately chose to compress the KV cache to 35%. This ensures POD to achieve performance comparable to LLaMA3-8B-32K using the same training data.

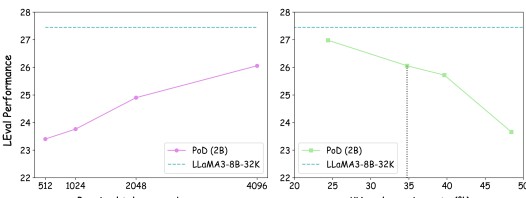

Figure 5: Relationship between model performance on LEval and two factors: proximal token count and KV cache savings rate

**Case study** In Figure 6, we provide four different representative cases to compare StreamingLLM, $H_2O$, and our POD. For case (a), The correct answer lies within the window of neighboring tokens, allowing all methods to attend to it during decoding. As a result, all three methods can make accurate predictions. For case (b), the answer is at the beginning, and both StreamingLLM and POD can attend to it during decoding, resulting in correct predictions. However, for $H_2O$, a long sequence of irrelevant tokens following the answer causes it to mistakenly discard the initial answer, leading to an incorrect prediction. For case (c), the answer is in the middle, and StreamingLLM cannot attend to it during decoding, leading to an incorrect prediction. However, the text following the answer and just before the question is related to the answer, allowing $H_2O$ to retain it within its attention window, resulting in a correct prediction. POD, being able to attend to all tokens, also makes a correct prediction. For case (d), only POD can answer the example in Needle in a Haystack since the other two methods disregard the answer tokens.

## 4 RELATED WORK

Long context LLMs present substantial challenges in terms of memory consumption and latency during inference, owing to their extensive parameter count and the long sequences they must process. From the perspective of optimization, we roughly categorize the approaches as follows:

---

*We run the official code.

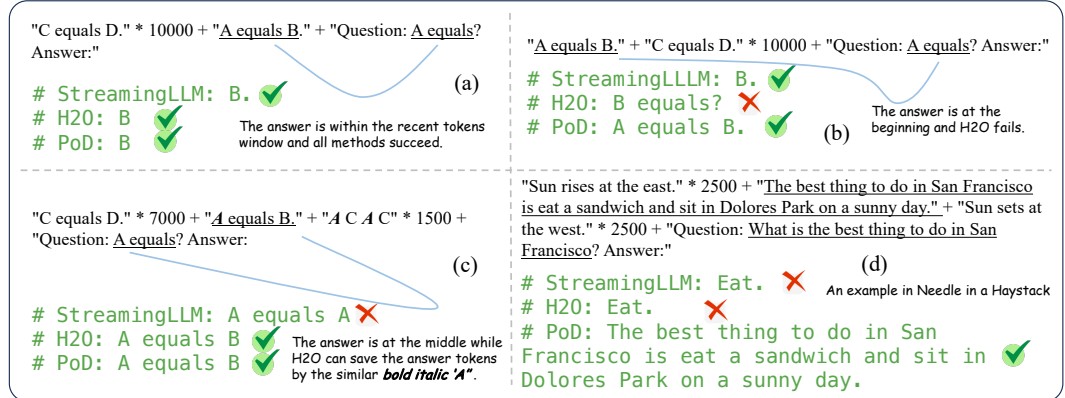

Figure 6: Case study of different methods. $s*n$ means repeating $n$ times of the string $s$. $+$ represents the concatenation of strings.

**Reduce context computation** Recent studies optimize context computation from two directions (Fu, 2024), i.e. prefilling and decoding. In the context of the prefilling phase, some work aims to reduce the size of the generated KV cache by selectively caching only significant input tokens like SnapKV (Li et al., 2024), PyramidKV (Zhang et al., 2024b), and LazyLLM (Fu et al., 2024). Meanwhile, MInference (Jiang et al., 2024a) and RetrievalAttention (Liu et al., 2024c) leverage the sparse attention mechanisms inherent in transformers to minimize prefilling latency. Some approaches (Li et al., 2023; Jiang et al., 2024b; Pan et al., 2024) also enhance efficiency by directly compressing the length of the input prompts. This work is orthogonal to these approaches and primarily focuses on optimizing the decoding phase.

In the context of the decoding phase, $H_2O$ (Zhang et al., 2023) drops insignificant tokens from the KV cache based on the attention scores, whereas LM-Infite (Han et al., 2024) and StreamingLLM (Xiao et al., 2024) retain only the most recent tokens along with several initial tokens during each decoding step. While all these methods enhance efficiency, they also carry the risk of discarding important tokens that are necessary for future text generation. This work aims to enhance efficiency while ensuring that no tokens are overlooked.

**Reduce hidden states dimension and quantize** We also present an alternative line of methods, including hidden size reduction and quantization, although these techniques are orthogonal to the focus of our work. On the one hand, MQA (Shazeer, 2019) and GQA (Ainslie et al., 2023) reduce the dimensionality of hidden states by grouping multiple heads of key-value pair states into a single pair. MLA (Liu et al., 2024a) compresses a pair of key-value states into a low-rank latent vector. On the other hand, AWQ (Lin et al., 2024) and QLLM (Liu et al., 2024d) convert model weights and activations into low bit-width formats, thereby reducing memory usage and computational overhead.

**Reduce redundancy between layers** Another line of work closely related to ours aims to improve efficiency by reducing redundancy between layers in transformers. LCKV (Wu & Tu, 2024), CLA (Brandon et al., 2024), and MiniCache (Liu et al., 2024b) exploit key-value similarities between layers by sharing key-value states across layers. In comparison to these approaches, our method has two main distinguishing features: 1) we leverage the similarity of attention scores between transformer layers (Xiao et al., 2019; Bhojanapalli et al., 2021) and scale the phenomenon to LLMs and 2) unlike their sharing strategies, which involve adjacent layers or the final layer, our sharing strategy is head-wise and derived from a search process.

## 5 CONCLUSION

In this work, we intend to improve the efficiency of LLMs. Previous window-based works suffer from performance degradation due to token loss. Thanks to the observation of proximal tokens are more important than distant tokens, we proposed PoD, which allocates fewer resources by sharing attention between similar layers for distant tokens. Evaluations reveal that PoD can save 35% of KV cache without sacrificing model performance. This approach not only optimizes resource allocation but also offers a pathway for future improvements in LLM efficiency.

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

## A  APPENDIX

For token $x_i$ at the $\ell$-th layer, we divide its context tokens into two groups: proximal tokens $T_P = \{j \mid x_j \text{ is a proximal token}\}$ and distant tokens $T_D = \{j \mid x_j \text{ is a distant token}\}$. The standard attention output to them is

$$
\begin{aligned}
\mathbf{o}_{\ell,i} &= \frac{\sum\limits_{j \in T_P \cup T_D} \exp \mathbf{a}_{\ell,i}^j \cdot \mathbf{V}_{\ell,j}}{\sum\limits_{j \in T_P \cup T_D} \exp \mathbf{a}_{\ell,i}^j} \\[2mm]
&= \frac{\sum\limits_{j \in T_P} \exp \mathbf{a}_{\ell,i}^j \cdot \mathbf{V}_{\ell,j}}{\sum\limits_{j \in T_P \cup T_D} \exp \mathbf{a}_{\ell,i}^j} + \frac{\sum\limits_{j \in T_D} \exp \mathbf{a}_{\ell,i}^j \cdot \mathbf{V}_{\ell,j}}{\sum\limits_{j \in T_P \cup T_D} \exp \mathbf{a}_{\ell,i}^j} \\[2mm]
&= \frac{\sum\limits_{j \in T_P} \exp \mathbf{a}_{\ell,i}^j}{\sum\limits_{j \in T_P \cup T_D} \exp \mathbf{a}_{\ell,i}^j} \cdot \frac{\sum\limits_{j \in T_P} \exp \mathbf{a}_{\ell,i}^j \cdot \mathbf{V}_{\ell,j}}{\sum\limits_{j \in T_P} \exp \mathbf{a}_{\ell,i}^j} + \frac{\sum\limits_{j \in T_D} \exp \mathbf{a}_{\ell,i}^j}{\sum\limits_{j \in T_P \cup T_D} \exp \mathbf{a}_{\ell,i}^j} \cdot \frac{\sum\limits_{j \in T_D} \exp \mathbf{a}_{\ell,i}^j \cdot \mathbf{V}_{\ell,j}}{\sum\limits_{j \in T_D} \exp \mathbf{a}_{\ell,i}^j} \\[2mm]
&= \frac{\sum\limits_{j \in T_P} \exp \mathbf{a}_{\ell,i}^j}{\sum\limits_{j \in T_P \cup T_D} \exp \mathbf{a}_{\ell,i}^j} \cdot \mathbf{o}_{\ell,i}^P + \frac{\sum\limits_{j \in T_D} \exp \mathbf{a}_{\ell,i}^j}{\sum\limits_{j \in T_P \cup T_D} \exp \mathbf{a}_{\ell,i}^j} \cdot \mathbf{o}_{\ell,i}^D .
\end{aligned} \tag{5}
$$

Therefore, we set

$$
g_{\ell,i} = \frac{\sum \exp \mathbf{a}_{\ell,i}^P}{\sum \exp \mathbf{a}_{\ell,i}^P + \sum \exp \mathbf{a}_{\ell,i}^D}. \tag{6}
$$

