# OpenReview forum: "Boosting Long-Context LLM Inference Efficiency with Intra-Layer Attention Similarity"
_ICLR.cc/2025/Conference — ICLR 2025 Conference Withdrawn Submission_

### Official Review · Reviewer_iXy9 · 2024-10-28

**Soundness:** 2
**Presentation:** 3
**Contribution:** 2
**Rating:** 3
**Confidence:** 3

**Summary:**

This paper introduces **POD** (Proximal Over Distant), a method to boost the efficiency of LLM inference with long contexts.

It addresses the memory and computation bottlenecks of the decoding by leveraging two insights: (1) Proximal Tokens Importance, where initial and recent tokens are more critical than distant tokens, allowing prioritized computation for these tokens; and (2) Intra-Layer Attention Similarity, where attention patterns between consecutive layers are similar, enabling shared attention scores for distant tokens across layers. POD optimizes inference by grouping layers based on this similarity, using shared attention for distant tokens, and minimizing memory usage in KV caching. Experiments show that POD can reduce KV cache memory by 35% while maintaining comparable accuracy to baseline models, effectively optimizing long-context LLMs without compromising on essential token information.

**Strengths:**

1. The paper creatively combines layer-similarity and post-trianing and proposes a novel approach **POD**, which improves inference efficiency by reducing memory requirements for less important distant tokens through inter-layer attention sharing, achieving a 35% reduction in KV cache while retaining all tokens for performance stability.

2. It provides evaluations across multiple long-context benchmarks and relevant ablation studies. **POD** outperforms baselines in most cases on provided experiments. In addition, the authors described the pipeline and details of **PoD** in a clear way.

**Weaknesses:**

1. The paper didn't explain the motivation in a reasonable way.

For example, (1) the observation 1 from Figure 1(a) is NOT really valid given the algorithm of StreamingLLM is to evict the tokens except for sink tokens (i.e. initial tokens) and most recent ones. In this case, "A equals B" inserted in the middle will be evicted and thus not used by StreamingLLM, which reasonably fails to return correct answer in the input. (2) The trend of similar attention scores shown in Figure 1(d) might not be immediately obvious to readers.

2. The setup of empirical results of the paper is relatively limited.

For example, (1) the paper only used LLaMA-3-8B model in experiment section, it would be better to how this approach performs when the LLM scales up; (2) the paper should also include some other SOTA token-selection-based approaches such as Quest: https://arxiv.org/abs/2406.10774

3. The training stage of **PoD** may add more computational costs the application of the approach.

**Questions:**

1. How does the approach apply to larger-scaled and up-to-date models such LLaMA-3-70B and LLaMA-3.1 models?

2. Does the approach outperform other token-selection-based sparse attention mechanisms other than pure eviction-based baselines?

3. Can authors justify the weakness 1. and elaborate the observation from Figure 1(a)?

---

> ### Author Response · Authors · 2024-11-25
> **Response to Reviewer iXy9**
>
> **Unreasonable motivation**
>
> - _Invalid observation 1_: We apologize for the misunderstanding **caused by a typo**. Indeed, Observation 1—"Proximal tokens (initial tokens + recent tokens) are more important than distant tokens"—is based on Figure 1-(c), not Figure 1-(a). Figure 1-(c) demonstrates that, even when attending to only the 256 proximal tokens, the model predicts the next token identically to the model attending to all tokens in 80% of the cases for the same input sequence. This phenomenon proves the observation 1. We have corrected the typo, which is highlighted in lines 80-81 of the revised version.
> - _Not obvious to the readers_: To help readers directly grasp the similarity of attention scores between consecutive layers, we made following two adjustments to Figure 1-(d):
>
>   - Replace it with a head that more clearly demonstrates the phenomenon
>   - Reduce the number of colors
>   - Distinguish only between values above and below the threshold (0.5)
>     We have updated Figure 1-(d) in the revised version.
>
> ---
>
> **Limited baselines** Please refer to Section **Response to all reviewers**.
>
> ---
>
> **Scaling of long context and other LLMs** Please refer to Section **Response to Reviewers K46j, jNu7 and iXy9**.
>
> ---
>
> **Scaling to LLaMA3-70B** Thank you for your valuable feedback and suggestions. We understand the importance of evaluating our approach on a more powerful model like LLaMA3-70B. However, due to the limited computational resources currently available to us, we are unable to conduct experiments on LLaMA3-70B within a short timeframe.
>
> To explore the generality of our method as thoroughly as possible, we have conducted experiments on two models, LLaMA3-8B and LLaMA3.1-8B, with two different context lengths (32K and 128K). We validated our approach on a variety of benchmarks, including Needle-in-a-Haystack, LongBench, LEval, and InfiniteBench.
>
> We hope this addresses your concern, but if you believe additional experiments on LLaMA3-70B are crucial, we are happy to include them in future work once the required resources become accessible.
>
> Thank you for your understanding and consideration.
>
> ---
>
> **Training cost of PoD** Please refer to Section **Response to Reviewers jNu7 and iXy9**.

---

> > ### Author Response · Authors · 2024-12-02
> > **Response to Reviewer iXy9**
> >
> > Dear Reviewer iXy9,
> >
> > Thank you sincerely for your thoughtful comments and suggestions on our paper. We have made every effort to address your concerns and hope the updates meet your expectations. In response, we have carefully revised our work and included additional experimental validations where necessary. We look forward to your feedback at your earliest convenience.
> >
> > Best regards,
> >
> > Team 13815

---

> > > ### Comment · Reviewer_iXy9 · 2024-12-02
> > >
> > > I would like to maintain the score. Thanks for the response.

---

> > > > ### Author Response · Authors · 2024-12-02
> > > > **Response to Reviewer iXy9**
> > > >
> > > > Dear Reviewer iXy9,
> > > >
> > > > Thank you for taking the time to reconsider our responses and for providing additional feedback. We greatly appreciate your effort and the detailed comments on our submission.
> > > >
> > > > We understand that your primary concerns lie in the following aspects:
> > > > - **Unclear motivation**: Two figures in the Introduction section could be clearer.
> > > > - **Limited baselines**: Absence of results on Quest.
> > > > - **Limited scale**: Need for scaling to longer contexts and other LLMs.
> > > >
> > > > To address these issues, we have taken the following actions:
> > > > - Resolved the typo issues and replotted clearer figures.
> > > > - Supplemented several baselines, including Quest.
> > > > - Conducted additional experiments on LLaMA3.1-8B with a 128K context window.
> > > >
> > > > We believe these updates specifically address the concerns you raised and hope this new information clarifies and strengthens our work. We are also grateful for your constructive suggestions, which have encouraged us to further refine our contributions to improving the efficiency of long-context LLMs. While we acknowledge that certain aspects (e.g., scaling to a 70B model) remain areas for improvement, we are confident that the revised submission addresses your key concerns to a reasonable extent and offers significant contributions to the field.
> > > >
> > > > We understand and respect your decision to maintain your current score. However, as the current score of 3 corresponds to a "reject," could you kindly provide more detailed feedback on the reasons for maintaining your current evaluation? Your insights would greatly help us identify areas for improvement and further refine our work.
> > > >
> > > > We remain deeply grateful for your thoughtful review and constructive feedback, and we sincerely value any additional insights you can share.
> > > >
> > > > Thank you once again for your time and valuable input.
> > > >
> > > > Best regards,
> > > >
> > > > Team 13815

---

### Official Review · Reviewer_jNu7 · 2024-10-31

**Soundness:** 2
**Presentation:** 3
**Contribution:** 2
**Rating:** 3
**Confidence:** 3

**Summary:**

The paper proposes the POD (Proximal over Distant) approach to improve inference efficiency in large language models (LLMs) by optimizing memory and computational resources for distant tokens instead of discarding them. It introduces a strategy of sharing attention scores across layers for distant tokens, based on the observation that proximal tokens (recent and initial tokens) are more crucial. By selectively reusing attention scores in deeper layers for distant tokens, POD achieves a 35% reduction in KV cache without significant performance loss. Extensive experiments demonstrate that POD maintains performance while enhancing efficiency across benchmarks.

**Strengths:**

The model demonstrates no performance drop with POD, and its speed is improved compared to dense methods.

**Weaknesses:**

1. The baseline methods for comparison are insufficient. Although the authors mention other acceleration methods in the Related Work section, these were not included in the experimental comparisons.
2. Compared to other training-free acceleration methods, this approach requires a continued training phase, adding computational cost.
3. Although not essential, it remains unclear how this method performs with longer context lengths, such as the 128k length achieved by LLaMA 3.1.

**Questions:**

I do not agree with the assumption that middle tokens are less important than edge tokens. This contradicts our prior knowledge, as the model may need to use information from any position during prediction. LM-Infinite and similar works have highlighted this issue, yet this is a limitation of the model rather than an optimization direction. Additionally, many papers achieve full accuracy in "needle-in-a-haystack" experiments, indicating that well-trained long-text models can overcome the "lost in the middle" issue.

---

> ### Author Response · Authors · 2024-11-25
> **Response to Reviewer jNu7**
>
> **Insufficient baselines** Please refer to Section **Response to all reviewers**.
>
> ---
>
> **Training cost of PoD** Please refer to Section **Response to Reviewers jNu7 and iXy9**.
>
> ---
>
> **Results over LLaMA3.1 with 128K context** Please refer to Section **Response to Reviewers K46j, jNu7 and iXy9**.
>
> ---
>
> **The discussion on whether edge tokens are more important than middle tokens** First, we acknowledge that, as you mentioned, the model may need information from any position during prediction. However, _the _**frequency**_ with which the model requires information from edge tokens versus middle tokens is significantly different_. Two empirical results support this point:
>
> 1. Experimental results in Figure 1-(c) of our manuscript demonstrates that, even when attending to only the 256 edge tokens, the model predicts the next token identically to the model attending to all tokens (32K) in **80%** of the cases for the same input sequence.
> 2. In Tables 1 and 2 of Section **Response to all reviewers (part 2)**, the model based on window attention maintains **80%-90%** of the performance of the dense model in practical tasks (Line 9 vs. Line 1).
>
> Therefore, we consider the importance of edge tokens and middle tokens from the perspective of the frequency with which the model requires their information, rather than their intrinsic importance for completing the task itself. More specifically, unlike token-eviction-based methods that directly discard middle tokens, we do not ignore these middle tokens. Instead, we reduce their impact on KV cache consumption through attention layer sharing, while ensuring that the model's performance is not significantly compromised. We hope this explanation alleviates your concerns.

---

> > ### Author Response · Authors · 2024-12-02
> > **Response to Reviewer jNu7**
> >
> > Dear Reviewer jNu7,
> >
> > Thank you sincerely for your thoughtful comments and suggestions on our paper. We have made every effort to address your concerns and hope the updates meet your expectations. In response, we have carefully revised our work and included additional experimental validations where necessary. We look forward to your feedback at your earliest convenience.
> >
> > Best regards,
> >
> > Team 13815

---

### Official Review · Reviewer_h78a · 2024-11-03

**Soundness:** 3
**Presentation:** 3
**Contribution:** 2
**Rating:** 8
**Confidence:** 3

**Summary:**

This paper proposes a method to reduce the size of KV cache to improve memory efficiency for long-context LLM. The proposed method involves sharing KV cache for “distant” tokens across layers, and continued pre-training the LM to adapt to this inference paradigm. Experiments are conducted with an extended LLaMA-3-8B model and evaluated on NIAH and two downstream tasks (LongBench and L-Eval). Results show that the proposed method saves 35% of KV cache while retaining performance, compared to previously proposed KV cache eviction method (H2O and StreamingLLM).

**Strengths:**

* This paper aims to improve efficiency for long-context language models, which is a practical and important problem.
* The paper is written clearly, with comprehensive analysis and experiments.

**Weaknesses:**

* **Baseline set-up**: The proposed method involves continuing pre-training the model to adapt to the paradigm of KV cache sharing across layers, yet all of the baselines (StreamingLLM, H2O) except for the “window attention”, are inference-time methods, making the comparison a bit unfair. It would be good to show what is the performance of adopting the proposed method as an inference-time method.
* **Experiment results**: The paper reported performance for all methods in Table 1 but only reported memory footprint of the proposed method (PoD) in table 2. What is the memory footprint saving for the baseline methods (StreamingLLM, etc.), and what is the performance-efficiency trade-off for different methods?

**Questions:**

The proposed method involves using the attention scores to set the gate for combining the attention output for the distant and proximal tokens. However, FlashAttention does not explicitly write out attention scores during the attention computation. I am curious how this is handled on the implementation side?

---

> ### Author Response · Authors · 2024-11-25
> **Response to Reviewer h78a (part 1)**
>
> **Baseline-setup** The additional lightweight post-training introduces a certain imbalance when comparing our method to token-eviction-based approaches (e.g., StreamingLLM, H2O). To address this, we could either apply post-training to these token-eviction-based methods or adapt our approach to be compatible with such methods. However, the former poses significant challenges, as the updated positional embeddings in StreamingLLM and the dynamic token eviction in H2O complicate training.  Conversely, adapting our method to token-eviction-based strategies conflicts with our primary objective of retaining all tokens. Therefore, to ensure a more equitable comparison, we have included additional baselines based on token-selection and layer-sharing, as outlined in the Section **Response to all reviewers**.

---

> ### Author Response · Authors · 2024-11-25
> **Response to Reviewer h78a (part 2)**
>
> **Performance-efficiency tradeoff** We supplement and compare the performance-efficiency tradeoff of all baselines and our methods in Table 4. We focus primarily on the compression of the KV cache, so in Table 4, we compare the different methods in terms of the optimized stage (prefilling or decoding), the proportion of KV cache saved, and the resulting performance loss. From this, we can draw the following conclusions:
>
> - Token-eviction-based methods (lines 4-8) are more efficient (suitable for both prefilling and decoding, and they save a relatively large amount of KV cache), but they incur a greater performance loss for the model.
> - Token-selection-based methods (SnapKV & PyramidKV, lines 1-2) can significantly compress the KV cache with minimal impact on model performance, but they are only applicable to the prefilling stage. Quest (line 3), on the other hand, does not degrade model performance and can be applied to both prefilling and decoding stages. However, it does not reduce the size of the KV cache; instead, it reduces the KV cache load (from the GPU's storage to the computation area) by restricting the model's attention to a selectively filtered subset of tokens.
> - Layer-selection-based methods can be applied to both the prefilling and decoding stages. Our method incurs much less performance degradation compared to the classical layer-sharing method, CLA (lines 9-10). Furthermore, _our method is orthogonal to token-selection-based methods_, and when combined (lines 1, 10, 11), the two approaches result in a model that excels in three aspects: optimized stage (applicable to both prefilling and decoding), KV cache saving (with a higher compression rate), and maintaining model performance.
>
> We have updated this in Table 3 and the highlighted lines 378-401 in the revised version.
>
> **Table 4: Performance-efficiency tradeoff comparison for LLaMA3-8B-32K [1-3: token-selection-based methods; 4-8: token-eviction-based methods; 9-11: layer-sharing-based methods]**
>
> | Order | Method                 | Optimized Stage       | KV Cache Saving (%) | Performance Degradation (%) |
> | ---------- | --------------------------- | -------------------------- | ------------------------ | -------------------------------- |
> | 1     | SnapKV                 | Prefilling            | 87.5                | 4.3                         |
> | 2     | PyramidKV              |        Prefilling               | 93.6                | 3.4                         |
> | 3     | Quest                  | Prefilling & Decoding | 0                   | 1.4                         |
> | 4     | LM-Infinite            |            Prefilling & Decoding           | 87.5                | 7.7                         |
> | 5     | StreamingLLM           |       Prefilling & Decoding                | 87.5                | 8.0                         |
> | 6     | H2O                    |            Prefilling & Decoding           | 87.5                | 7.4                         |
> | 7     | Window Attention       |        Prefilling & Decoding               | 87.5                | 65.9                        |
> | 8     | Window Attention + CPT |        Prefilling & Decoding               | 87.5                | 12.6                        |
> | 9     | CLA                    |            Prefilling & Decoding           | 50.0                | 31.4                        |
> | 10    | PoD (ours)             |            Prefilling & Decoding           | 35.0                | 2.8                         |
> | 11    | PoD+SnapKV(ours)       |         Prefilling & Decoding              | 91.9                | 3.1                         |
>
> ---
>
> **Implementation of the gate** As demonstrated in Formulas (3) and (4) on page 5 of our manuscript, calculating the gate involves obtaining the **exponential sum** across each row of $QK^T$(with dimensions of _batch size × head number × sequence length_), rather than itself (_with dimensions of batch size x head number x sequence length x sequence length_). As you noted, FlashAttention does not return the complete matrix but outputs the exponential sum across each row [1].
>
> We will release our code in the future, which will include more implementation details.
>
> [1] [https://github.com/Dao-AILab/flash-attention/blob/v2.5.7/flash_attn/flash_attn_interface.py#L824](https://github.com/Dao-AILab/flash-attention/blob/v2.5.7/flash_attn/flash_attn_interface.py#L824)

---

> > ### Comment · Reviewer_h78a · 2024-11-27
> > **Reply to author's response**
> >
> > Thank you for the response and for the additional baselines and experiments!
> >
> > I have two further questions:
> > * Is the CLA baseline applied as an inference-time method or does it involve post-training the model too?
> > * As mentioned in my initial review, PoD can also be applied as an inference-time method (just like the eviction-based methods), i.e. sharing attention scores across layers. What's the performance of that? I think it is reasonable that it performs worse than the model with post training, or the other eviction-based methods, given that the post-training is lightweight. Yet, I think it will be valuable to include it as an analysis (also, as it is the motivation for the method).

---

> > > ### Author Response · Authors · 2024-11-28
> > > **Response to the reviewer**
> > >
> > > Thank you for your further response. Here are our answers:
> > > * CLA also requires post-training. Specifically, to ensure fairness, we used **exactly the same data** to train both the CLA and our model in our experiments.
> > > * Yes, your suggestion makes our work more complete. We have supplemented the results as follows:
> > >
> > > **Table 1: Evaluation results of different methods on LongBench [1: dense; 2-4: token-selection-based methods; 5-9: token-eviction-based methods; 10-13: layer-sharing-based methods]**
> > >
> > > | Order | Model                  | Window      | Avg.  | SQA   | MQA   | Summ  | Few-Shot | Code  |
> > > | ---------- | --------------------------- | ---------------- | ---------- | ---------- | ---------- | ---------- | ------------- | ---------- |
> > > | 1     | LLaMA3-8B-32K          | 32K         | 45.28 | 32.94 | 32.23 | 25.41 | 69.30    | 66.54 |
> > > | 2     | SnapKV                 | 4K          | 44.17 | 31.76 | 31.85 | 21.92 | 68.62    | 66.72 |
> > > | 3     | PyramidKV              | 4K          | 44.78 | 33.34 | 31.51 | 23.76 | 68.91    | 66.36 |
> > > | 4     | Quest                  | 4K          | 44.82 | 32.14 | 32.19 | 24.27 | 69.05    | 66.43 |
> > > | 5     | LM-Infinite            | 4+4092      | 42.84 | 28.83 | 28.95 | 21.74 | 68.12    | 66.54 |
> > > | 6     | StreamingLLM           | 4+4092      | 42.80 | 28.68 | 28.95 | 21.64 | 68.14    | 66.6  |
> > > | 7     | H2O                    | 96+4000     | 43.24 | 29.36 | 29.51 | 22.73 | 68.45    | 66.17 |
> > > | 8     | Window Attention       | 4K          | 14.76 | 8.90  | 3.63  | 9.05  | 11.13    | 41.08 |
> > > | 9     | Window Attention + CPT | 4K          | 41.98 | 26.94 | 27.95 | 22.29 | 66.60    | 66.10 |
> > > | 10    | CLA                    | 32K         | 37.87 | 24.02 | 22.58 | 22.50 | 60.92    | 59.35 |
> > > | 11    | PoD w/o CPT (ours)| 16+4080+28K| 40.87| 27.15| 26.44|20.73|64.50|65.51|
> > > | 12    | PoD (ours)             | 16+4080+28K | 44.76 | 30.97 | 32.43 | 24.82 | 67.30    | 68.26 |
> > > | 13    | PoD+SnapKV(ours)       | 4K          | 44.25 | 30.98 | 32.68 | 22.90 | 66.90    | 67.79 |
> > >
> > > ---
> > >
> > > **Table 2: Evaluation results of different methods on LEval [1: dense; 2-4: token-selection-based methods; 5-9: token-eviction-based methods; 10-13: layer-sharing-based methods]**
> > >
> > > | Order | Model                  | Window      | Avg.  | Closed | QA    | Summ  |
> > > | ---------- | --------------------------- | ---------------- | ---------- | ----------- | ---------- | ---------- |
> > > | 1     | LLaMA3-8B-32K          | 32K         | 27.44 | 42.10  | 24.68 | 15.55 |
> > > | 2     | SnapKV                 | 4K          | 25.76 | 39.86  | 23.90 | 13.53 |
> > > | 3     | PyramidKV              | 4K          | 25.90 | 42.10  | 22.63 | 12.96 |
> > > | 4     | Quest                  | 4K          | 26.94 | 40.55  | 25.59 | 14.69 |
> > > | 5     | LM-Infinite            | 4+4092      | 24.68 | 37.32  | 22.8  | 13.91 |
> > > | 6     | StreamingLLM           | 4+4092      | 24.57 | 37.12  | 22.79 | 13.81 |
> > > | 7     | H2O                    | 96+4000     | 24.60 | 37.15  | 23.18 | 13.48 |
> > > | 8     | Window Attention       | 4K          | 9.77  | 20.95  | 5.57  | 2.79  |
> > > | 9     | Window Attention + CPT | 4K          | 22.53 | 32.94  | 22.09 | 12.55 |
> > > | 10    | CLA                    | 32K         | 14.70 | 19.05  | 13.52 | 11.52 |
> > > | 11    | PoD w/o CPT (ours)  | 16+4080+28K | 23.39 | 37.45 | 20.43 | 12.29|
> > > | 12    | PoD (ours)             | 16+4080+28K | 27.18 | 43.59  | 22.95 | 15.00 |
> > > | 13    | PoD+SnapKV(ours)       | 4K          | 26.50 | 43.07  | 22.12 | 14.32 |
> > >
> > > From the two tables above, we can confirm that without post-training, the performance of PoD indeed declines. However, with lightweight training on a small amount of data, its performance recovers to a level comparable to that of dense models.
> > >
> > > We sincerely hope that we have resolved your issue and look forward to your reply. Thank you very much!

---

> > > > ### Comment · Reviewer_h78a · 2024-11-28
> > > > **Thank you for your reply**
> > > >
> > > > Thank you for the reply! The author's response has addressed my concerns, and I have changed my score from 6 to 8.

---

### Official Review · Reviewer_K46j · 2024-11-04

**Soundness:** 2
**Presentation:** 3
**Contribution:** 2
**Rating:** 5
**Confidence:** 4

**Summary:**

This paper proposes POD, a method to improve LLM inference efficiency by differentiating between proximal tokens (recent and initial tokens) and distant tokens. Instead of discarding tokens like previous approaches, POD shares attention scores across layers for distant tokens while keeping full computation for proximal tokens. The authors show their method can save 35% KV cache without significant performance degradation on tasks like LongBench and LEval.

**Strengths:**

- The work studies an important problem in LLM deployment, as handling long contexts efficiently is becoming increasingly crucial.
- The method is relatively straightforward to implement and requires minimal adaptation of existing models. As shown in Section 2.2, it mainly involves grouping similar layers and modifying the attention computation, without needing extensive finetuning or architectural changes.

**Weaknesses:**

- The paper misses comparison with some important recent baselines, particularly SnapKV [1] and PyramidKV [2], which also address KV cache optimization. While the related work section mentions them briefly, not including them in the experimental comparison makes it difficult to assess the relative advantages of POD.
- The evaluation seems limited in scope given the current state of the field. The authors only test on LLaMA3-8B with 32K context, while recent models routinely handle 128K tokens, e.g., Llama 3.1 and 3.2, Mistral Nemo, Phi 3.5, and so on. This raises questions about how POD would scale to longer contexts and whether the benefits would hold at larger scales.
- Also, the methods were only evaluated on LLaMA3-8B. It is unknown whether the method only works for this specific model or whether the method can be generally applied to most LLMs.
- The implementation details raise concerns about compatibility with modern attention optimizations. The head-wise different grouping of layers (shown in Figure 2) suggests that each attention head would need different attention patterns, which may make it incompatible with efficient implementations like FlashAttention. Have you explored the compatibility of POD with FlashAttention or similar optimized attention implementations?
- The memory savings claims could be better substantiated. While the paper reports 35% KV cache savings, Table 2 shows somewhat inconsistent practical gains across different batch sizes, and there's limited analysis of the overhead introduced by maintaining separate attention patterns per head. Could you clarify the computational overhead of maintaining different attention patterns for each head?

[1] Yuhong Li, Yingbing Huang, Bowen Yang, Bharat Venkitesh, Acyr Locatelli, Hanchen Ye, Tianle Cai, Patrick Lewis, and Deming Chen. Snapkv: Llm knows what you are looking for before generation. arXiv preprint arXiv:2404.14469, 2024.
[2] Yichi Zhang, Bofei Gao, Tianyu Liu, Keming Lu, Wayne Xiong, Yue Dong, Baobao Chang, Junjie Hu, Wen Xiao, et al. Pyramidkv: Dynamic kv cache compression based on pyramidal information funneling. arXiv preprint arXiv:2406.02069, 2024.

**Questions:**

See the Weaknesses above.

---

> ### Author Response · Authors · 2024-11-25
> **Response to Reviewer K46j**
>
> **Missing baselines** Please refer to Section **Response to all reviewers**.
>
> ---
>
> **Longer context and other LLMs** Please refer to Section **Response to Reviewers K46j , jNu7 and iXy9**.
>
> ---
>
> **Compatibility with modern attention optimizations**
>
> - First, sorry for the misunderstanding.
>
> > The attention pattern for each attention head is the _**same**_. The difference lies in the layers that _**share the attention scores**_ for distant tokens.
>
> For example, in Figure 2-①, all heads use the attention mask (pattern) shown in Figure 2-②. However, for head 1, the attention scores for distant tokens in layer 3 are derived from the results computed in layer 2. In contrast, for head H, the attention scores for distant tokens in layer 3 are derived from the results computed in layer 1.
>
> - Second, the implementation is fully compatible with modern attention optimizations. As described in Section 2.2 of our manuscript, the computation for each token in the attention module is divided into two parts: computation for proximal tokens and distant tokens. Let $Q_{l}^h, K_{l}^h, V_l^h$ denote the query, key, and values states of the $l\text{-th}$ layer for head $h$. $H$ is the number of heads.
>   - For proximal tokens, there is no sharing of information across layers, and the attention pattern follows the approach of StreamingLLM [1]. Since FlashAttention does not currently support this, we utilized PyTorch's FlexAttention [2], an efficient attention implementation technique that supports custom masks, i.e.,
>
>     $O_l^P = \text{FlexAttention}\left(\text{Concat}\left( \left\\{Q_l^h\right\\}_{1\le h\le H} \right)  ,  \text{Concat}\left( \left\\{ K_l^h \right \\} _{1\le h\le H} \right)  , \text{Concat}\left( \left \\{ V_l^h \right \\} _{ 1\le h\le H } \right) \right)$ (1)
>
>
>   - For distant tokens, different heads may share attention scores with different layers. Since most modern optimized attention mechanisms do not support returning the attention scores (with dimensions: batch size × head number × sequence length × sequence length), we pass the query and key states from the lower layer to the upper layer for each head. The upper layer then concatenates these head-specific states and recalculates the attention, i.e.,
>
>     $\\tilde{Q} _l^h=\\begin{cases} Q _l^h, & \text{if no sharing between layer }l \text{ and } l-1 , \\\ Q _{l-1}^h, & \text{otherwise} . \\end{cases}$  (2)
>
>     $\tilde{K} _l^h=\begin{cases} K _l^h, & \text{if no sharing between layer }l \text{ and } l-1 , \\\ K _{l-1}^h, & \text{otherwise} . \end{cases}$ (3)
>
>      $O_l^D = \text{FlexAttention}\left(\text{Concat}\left( \left\\{\\tilde{Q} _l^h\right\\} _{1\le h\le H} \right)  ,  \text{Concat}\left( \left\\{ \tilde{K} _l^h \right \\} _{1\le h\le H} \right)  , \text{Concat}\left( \left \\{ V_l^h \right \\} _{ 1\le h\le H } \right) \right)$ (4)
>
> We will release our code in the future, which will include more implementation details.
>
> ---
>
> **Computational overhead of managing distinct attention patterns for each head** As mentioned above, the attention pattern for each head is the same. Compared to standard dense attention, the computational overhead of our method primarily comes from the transmission of query states and key states between layers. During inference, the key states come from the KV cache, so only the transmission of query states between layers adds extra computational overhead.
>
> In the prefilling phase, the main memory usage comes from the calculation of attention scores (_batch size x number of heads x sequence length x sequence length_) (with Flash Attention optimizing this part). The query states transmitted between layers have dimensions of (_batch size x number of heads x sequence length x model dimension_), which is much smaller compared to the attention score when the context is long. Similarly, in the decoding phase, only one token's query state is transmitted between layers at each step, which is minimal compared to the key and value states of all tokens already cached in the KV cache. Therefore, the computational overhead introduced by our method is relatively small.
>
> ---
>
> [1] Xiao, Guangxuan, et al. "Efficient streaming language models with attention sinks." _arXiv preprint arXiv:2309.17453_ (2023).
>
> [2] [https://pytorch.org/blog/flexattention/](https://pytorch.org/blog/flexattention/)

---

> > ### Author Response · Authors · 2024-12-02
> > **Response to Reviewer K46j**
> >
> > Dear Reviewer K46j,
> >
> > Thank you sincerely for your thoughtful comments and suggestions on our paper. We have made every effort to address your concerns and hope the updates meet your expectations. In response, we have carefully revised our work and included additional experimental validations where necessary. We look forward to your feedback at your earliest convenience.
> >
> > Best regards,
> >
> > Team 13815

---

### Author Response · Authors · 2024-11-25
**Response to all reviewers (Part 2)**

**Table 1: Evaluation results of different methods on LongBench [1: dense; 2-4: token-selection-based methods; 5-9: token-eviction-based methods; 10-12: layer-sharing-based methods]**

| Order | Model                  | Window      | Avg.  | SQA   | MQA   | Summ  | Few-Shot | Code  |
| ---------- | --------------------------- | ---------------- | ---------- | ---------- | ---------- | ---------- | ------------- | ---------- |
| 1     | LLaMA3-8B-32K          | 32K         | 45.28 | 32.94 | 32.23 | 25.41 | 69.30    | 66.54 |
| 2     | SnapKV                 | 4K          | 44.17 | 31.76 | 31.85 | 21.92 | 68.62    | 66.72 |
| 3     | PyramidKV              | 4K          | 44.78 | 33.34 | 31.51 | 23.76 | 68.91    | 66.36 |
| 4     | Quest                  | 4K          | 44.82 | 32.14 | 32.19 | 24.27 | 69.05    | 66.43 |
| 5     | LM-Infinite            | 4+4092      | 42.84 | 28.83 | 28.95 | 21.74 | 68.12    | 66.54 |
| 6     | StreamingLLM           | 4+4092      | 42.80 | 28.68 | 28.95 | 21.64 | 68.14    | 66.6  |
| 7     | H2O                    | 96+4000     | 43.24 | 29.36 | 29.51 | 22.73 | 68.45    | 66.17 |
| 8     | Window Attention       | 4K          | 14.76 | 8.90  | 3.63  | 9.05  | 11.13    | 41.08 |
| 9     | Window Attention + CPT | 4K          | 41.98 | 26.94 | 27.95 | 22.29 | 66.60    | 66.10 |
| 10    | CLA                    | 32K         | 37.87 | 24.02 | 22.58 | 22.50 | 60.92    | 59.35 |
| 11    | PoD (ours)             | 16+4080+28K | 44.76 | 30.97 | 32.43 | 24.82 | 67.30    | 68.26 |
| 12    | PoD+SnapKV(ours)       | 4K          | 44.25 | 30.98 | 32.68 | 22.90 | 66.90    | 67.79 |

---

**Table 2: Evaluation results of different methods on LEval [1: dense; 2-4: token-selection-based methods; 5-9: token-eviction-based methods; 10-12: layer-sharing-based methods]**

| Order | Model                  | Window      | Avg.  | Closed | QA    | Summ  |
| ---------- | --------------------------- | ---------------- | ---------- | ----------- | ---------- | ---------- |
| 1     | LLaMA3-8B-32K          | 32K         | 27.44 | 42.10  | 24.68 | 15.55 |
| 2     | SnapKV                 | 4K          | 25.76 | 39.86  | 23.90 | 13.53 |
| 3     | PyramidKV              | 4K          | 25.90 | 42.10  | 22.63 | 12.96 |
| 4     | Quest                  | 4K          | 26.94 | 40.55  | 25.59 | 14.69 |
| 5     | LM-Infinite            | 4+4092      | 24.68 | 37.32  | 22.8  | 13.91 |
| 6     | StreamingLLM           | 4+4092      | 24.57 | 37.12  | 22.79 | 13.81 |
| 7     | H2O                    | 96+4000     | 24.60 | 37.15  | 23.18 | 13.48 |
| 8     | Window Attention       | 4K          | 9.77  | 20.95  | 5.57  | 2.79  |
| 9     | Window Attention + CPT | 4K          | 22.53 | 32.94  | 22.09 | 12.55 |
| 10    | CLA                    | 32K         | 14.70 | 19.05  | 13.52 | 11.52 |
| 11    | PoD (ours)             | 16+4080+28K | 27.18 | 43.59  | 22.95 | 15.00 |
| 12    | PoD+SnapKV(ours)       | 4K          | 26.50 | 43.07  | 22.12 | 14.32 |

---

### Author Response · Authors · 2024-11-25
**Response to all reviewers (Part 1)**

Thank you to all the reviewers for the effort you put into this work. Your suggestions have truly helped improve our work.

One of the concerns you raised was **the lack of baselines and unclear comparisons**. In response, we have further included the baselines you suggested and provided a detailed discussion between these baselines and our approach. In particular, we categorize these methods into three groups:

1. **Token-selection-based methods**: SnapKV [1], PyramidKV [2], Quest [3]
2. **Token-eviction-based methods**: LM-Infinite [4], StreamingLLM [5], H2O [6], Window Attention [7]
3. **Layer-sharing-based methods**: CLA [8], PoD (ours)

We have supplemented the results in Tables 1 and 2 (**part 2**). From these, we can conclude the following:

- Token-eviction-based methods (Lines 5-9 vs. Line 1) degrade performance.
- Both token-selection-based methods (Lines 2-4 vs. Line 1) and our method (PoD, a layer-sharing-based method, Line 11 vs. Line 1) maintain performance. Furthermore, _our method is orthogonal to token-selection-based methods_. Specifically, by integrating the typical token-selection-based method SnapKV with our PoD, we obtain a more efficient model (saving KV cache at both token and layer levels) that maintains performance (Line 12 vs. Line 1).
- Our method outperforms the classical layer-sharing-based method CLA (Line 11 vs. Line 10). When post-trained with the same amount of data, CLA significantly degrades performance (Line 10 vs. Line 1). In the original work, CLA was pretrained from scratch. We attribute the performance degradation to the fact that a pretrained LLM cannot effectively adapt to CLA with a relatively small amount of data.

We have updated the results in Table 1, as well as the comparison of all methods, which can be found in the highlighted lines 297-315 and 355-362 of the revised version.

---

[1] Li, Yuhong, et al. "Snapkv: Llm knows what you are looking for before generation." _arXiv preprint arXiv:2404.14469_ (2024).

[2] Cai, Zefan, et al. "Pyramidkv: Dynamic kv cache compression based on pyramidal information funneling." _arXiv preprint arXiv:2406.02069_ (2024).

[3] Tang, Jiaming, et al. "Quest: Query-Aware Sparsity for Efficient Long-Context LLM Inference." _arXiv preprint arXiv:2406.10774_ (2024).

[4] Han, Chi, et al. "LM-Infinite: Zero-Shot Extreme Length Generalization for Large Language Models." _Proceedings of the 2024 Conference of the North American Chapter of the Association for Computational Linguistics: Human Language Technologies (Volume 1: Long Papers)_. 2024.

[5] Xiao, Guangxuan, et al. "Efficient streaming language models with attention sinks." _arXiv preprint arXiv:2309.17453_ (2023).

[6] Zhang, Zhenyu, et al. "H2o: Heavy-hitter oracle for efficient generative inference of large language models." _Advances in Neural Information Processing Systems_ 36 (2023): 34661-34710.

[7] Beltagy, Iz, Matthew E. Peters, and Arman Cohan. "Longformer: The long-document transformer." _arXiv preprint arXiv:2004.05150_ (2020).

[8] Brandon, William, et al. "Reducing Transformer Key-Value Cache Size with Cross-Layer Attention." _arXiv preprint arXiv:2405.12981_ (2024).

---

### Author Response · Authors · 2024-11-25
**Response to Reviewers K46j, jNu7 and iXy9 (part 2)**

**Table 3: Evaluation results of different methods over 7 sub-tasks in the InfiniteBench under different context sizes** **[1: dense; 2-4: token-selection-based methods; 5-9: token-eviction-based methods; 10-12: layer-sharing-based methods]. OOM: out of memory over one A800-80G GPU**

| Order | Model                  | Window          | Avg.  | 32K   |       |        |           |            |              |             | 64K   |       |        |           |            |              |             | 128K  |       |        |           |            |              |             |
|-------|------------------------|-----------------|-------|-------|-------|--------|-----------|------------|--------------|-------------|-------|-------|--------|-----------|------------|--------------|-------------|-------|-------|--------|-----------|------------|--------------|-------------|
|       |                        |                 |       | En.QA | En.MC | En.Sum | Math.Find | Code.Debug | Retr.PassKey | Retr.Number | En.QA | En.MC | En.Sum | Math.Find | Code.Debug | Retr.PassKey | Retr.Number | En.QA | En.MC | En.Sum | Math.Find | Code.Debug | Retr.PassKey | Retr.Number |
| 1     | LLaMA3.1-8B-128K       | 128K            | 36.37 | 29.56 | 25.33 | 15.51  | 32.29     | 27.41      | 27.12        | 27.12       | 37.26 | 28.82 | 15.14  | 28.00     | 27.92      | 54.24        | 54.24       | 31.24 | 41.78 | 14.86  | 21.14     | 25.38      | 100.00       | 99.49       |
| 2     | SnapKV                 | 4K              | 35.43 | 28.93 | 26.20 | 13.26  | 32.28     | 27.41      | 27.12        | 26.61       | 35.14 | 27.95 | 11.69  | 28.00     | 27.92      | 54.24        | 53.05       | 29.34 | 34.50 | 13.65  | 21.14     | 26.65      | 100.00       | 98.98       |
| 3     | PyramidKV              | 4K              | OOM*   | 29.88 | 26.20 | 15.01  | 32.29     | 27.39      | 27.12        | 27.12       | 36.53 | 29.26 | 15.18  | 28.00     | 27.92      | 54.24        | 54.24       | OOM   |       |        |           |            |              |             |
| 4     | Quest                  | 4K              | 35.17 | 28.49 | 25.76 | 12.87  | 32.29     | 27.41      | 27.12        | 27.12       | 35.81 | 27.51 | 11.22  | 27.89     | 27.90      | 54.24        | 54.24       | 29.04 | 34.50 | 8.95   | 21.14     | 26.90      | 100.00       | 98.14       |
| 5     | LM-Infinite            | 4+4092          | 18.08 | 25.28 | 26.64 | 12.46  | 32.11     | 27.31      | 3.39         | 3.39        | 27.25 | 28.82 | 12.64  | 26.89     | 27.91      | 3.39         | 3.39        | 22.19 | 29.26 | 13.01  | 21.14     | 26.63      | 3.39         | 3.22        |
| 6     | StreamingLLM           | 4+4092          | 18.05 | 25.39 | 26.20 | 12.05  | 32.29     | 27.41      | 3.39         | 3.39        | 27.14 | 27.95 | 13.29  | 28.00     | 27.92      | 3.39         | 3.39        | 22.66 | 27.95 | 12.79  | 21.14     | 26.65      | 3.39         | 3.22        |
| 7     | H2O                    | 96+4000         | OOM*   | 25.42 | 26.20 | 13.08  | 32.29     | 27.41      | 4.92         | 3.39        | 27.55 | 27.95 | 14.31  | 28.00     | 27.91      | 7.12         | 3.56        | OOM   |       |        |           |            |              |             |
| 8     | Window Attention       | 4K              | 2.69  | 3.52  | 3.49  | 0.53   | 0.00      | 10.41      | 0.00         | 1.36        | 3.36  | 3.06  | 0.65   | 0.00      | 10.66      | 0.00         | 1.19        | 3.32  | 3.49  | 0.70   | 0.00      | 9.64       | 0.00         | 1.19        |
| 9     | Window Attention + CPT | 4K              | 13.77 | 12.59 | 18.78 | 11.31  | 18.57     | 28.43      | 3.39         | 3.39        | 13.12 | 17.90 | 10.78  | 17.71     | 28.68      | 3.39         | 3.39        | 12.81 | 20.96 | 11.26  | 17.71     | 28.17      | 3.39         | 3.39        |
| 10    | CLA                    | 32K             | 32.20 | 22.07 | 34.06 | 13.58  | 12.57     | 27.16      | 24.07        | 25.76       | 22.74 | 31.88 | 12.74  | 12.57     | 27.66      | 50.85        | 52.54       | 21.62 | 34.50 | 12.96  | 12.86     | 29.44      | 97.80        | 96.78       |
| 11    | PoD (ours)             | 16+4080+28K     | 36.22 | 27.43 | 38.43 | 17.89  | 23.68     | 25.61      | 26.61        | 27.12       | 29.62 | 41.92 | 15.52  | 20.78     | 26.29      | 53.73        | 54.24       | 26.64 | 42.79 | 15.52  | 22.16     | 25.76      | 99.83        | 99.15       |
| 12    | PoD+SnapKV (ours)      | 16+4080+28K; 4K | 34.77 | 28.61 | 38.86 | 13.90  | 23.68     | 25.61      | 26.61        | 23.39       | 28.48 | 43.23 | 14.03  | 20.78     | 26.29      | 53.73        | 49.49       | 24.72 | 40.61 | 12.36  | 22.16     | 25.76      | 99.83        | 87.97       |


\* ***We run the official code.***

---

> ### Comment · Reviewer_jNu7 · 2024-11-25
> **Response to Authors**
>
> Thank you for providing responses.
>
> InfiniteBench comprises 12 unique tasks. Why are there results for only four tasks, especially omitting the widely used Longbook QA? Additionally, the HELMET benchmark mentioned in reference [1] includes a wider variety of tasks and might be a better choice for testing. I also noticed that the Longbench results missed the Synthetic task.

---

> > ### Author Response · Authors · 2024-12-02
> > **Response to Reviewer jNu7**
> >
> > Apologies for the delayed response as our experiments require a long training process. Over the past few days, we have done our best to address your concerns as follows:
> > - We have increased the number of tasks evaluated on InfiniteBench to 7 (*including LongBookQA*). The results have been updated in Table 3 in Section **Response to Reviewers K46j, jNu7 and iXy9 (part 2)**. The reasons for not evaluating the remaining 5 tasks are as follows:
> >   - longbook_qa_chn: This task focuses on Chinese, whereas our work only considers English.
> >   - LLaMA3.1-8B performs poorly on tasks such as kv_retrieval, math_calc, code_run, and longdialogue_qa_eng (below 10% or even 5%), making it less meaningful to evaluate these tasks.
> >
> > Unfortunately, we found that the previous version of our model performed poorly on synthetic tasks (passkey and number_string), trailing the dense model by 30 points. Upon analysis, we discovered that this was because the data formats involved in constructing these synthetic tasks were almost absent in our post-training dataset.
> > For example, the passkey task requires the model to retrieve numeric strings hidden within large text passages. However, since our post-training data contained almost no numeric strings, the model consistently failed in its predictions.
> >
> > To address this issue, we implemented a remedy: *we incorporated a very small amount (100 samples, approximately 0.2% of the fine-tuning data) of such synthetic data into our post-training dataset. **These samples do not overlap with the evaluation data.*** As a result, we observed that the performance of our model on these tasks became comparable to that of the dense model (Retr.PassKey and Retr.Number in Table 3). Since the construction of synthetic data on LongBench is relatively indirect (requiring text rewriting using GPT-4), we conducted experimental validation only on the synthetic tasks from InfiniteBench.
> >
> > Finally, we commit to updating this limitation in the subsequent version. Thank you for your suggestion.
> > - HELMET is indeed a great benchmark for evaluating large language models' ability to handle long contexts. However, it focuses more on evaluating much longer contexts (e.g., with a maximum length of up to 5M, and most tasks reaching over 800K). Its scale exceeds the scope of what we are currently discussing. In the future, when we work on larger models with longer contexts, we will validate them on this benchmark.

---

### Author Response · Authors · 2024-11-25
**Response to Reviewers K46j, jNu7 and iXy9 (part 1)**

**128K context over LLaMA3.1-8B** To explore the generality of our method, we conducted experiments on LLaMA3.1-8B, which can handle longer (128K) contexts. We sampled 5B tokens from the ProLong-data-512K [1] dataset and applied the same hyperparameter configuration used for training LLaMA3-8B-32K to post-train LLaMA3.1-8B with a sequence length of 128K. The evaluation results over 4 practical sub-tasks in the InfiniteBench [2] under different context sizes are shown in the following Table 3 (**part 2**).

Consistent with the conclusions found in Tables 1 and 2 (Section **Response to all reviewers (part 2)**), our method causes less performance degradation compared to token-eviction-based methods. However, a notable difference is that token-selection-based methods appear to struggle in maintaining model performance in longer context scenarios. This limitation is also reflected in the combined model (PoD+SnapKV), which integrates our method with token-selection-based methods, showing a decline in performance. This to some extent indicates that our method is more robust to the context length.

We have updated the results in Table 4, as well as the conclusion, which can be found in the highlighted lines 422-446 of the revised version.

[1] Gao, Tianyu, et al. "How to train long-context language models (effectively)." _arXiv preprint arXiv:2410.02660_ (2024).

[2] Zhang, Xinrong, et al. "∞ Bench: Extending long context evaluation beyond 100k tokens." _Proceedings of the 62nd Annual Meeting of the Association for Computational Linguistics (Volume 1: Long Papers)_. 2024.

---

### Author Response · Authors · 2024-11-25
**Response to Reviewers jNu7 and iXy9**

**Training cost of PoD** To adapt a pre-trained LLM to PoD, additional post-training is indeed required. However, this post-training is very lightweight and efficient, as evidenced by two key aspects:

1. The amount of data required for post-training (5B tokens) constitutes **less than 0.04%** of the pretraining data (over 15T tokens).
2. Compared to the classical layer-sharing-based method CLA, our method PoD can restore performance comparable to the dense model using only 5B tokens, whereas CLA's performance still lags significantly behind the dense model (Lines 1, 10, and 11 of Tables 1 and 2 in Section **Response to all reviewers (part 2)**).

---

### Note · Authors · 2024-12-13

**Comment:**

We sincerely thank all the reviewers and relevant staff for their contributions to improving our work.

**Withdrawal Confirmation:**

I have read and agree with the venue's withdrawal policy on behalf of myself and my co-authors.